# PLA2G2A Phospholipase Promotes Fatty Acid Synthesis and Energy Metabolism in Pancreatic Cancer Cells with *K-ras* Mutation

**DOI:** 10.3390/ijms231911721

**Published:** 2022-10-03

**Authors:** Mingquan Zhang, Rong Xiang, Christophe Glorieux, Peng Huang

**Affiliations:** 1State Key Laboratory of Oncology in Southern China, Sun Yat-Sen University Cancer Center, Collaborative Innovation Center for Cancer Medicine, Guangzhou 510060, China; 2Metabolic Innovation Center, Sun Yat-sen University, Guangzhou 510060, China

**Keywords:** K-ras, PLA2G2A, tanshinone I, phospholipase, mitochondria, fatty acid synthesis, lipid metabolism, pancreatic cancer

## Abstract

Oncogenic *K-ras* is often activated in pancreatic ductal adenocarcinoma (PDAC) due to frequent mutation (>90%), which drives multiple cellular processes, including alterations in lipid metabolism associated with a malignant phenotype. However, the role and mechanism of the altered lipid metabolism in K-ras-driven cancer remains poorly understood. In this study, using human pancreatic epithelial cells harboring inducible K-ras^G12D^ (HPNE/*K-ras*^G12D^) and pancreatic cancer cell lines, we found that the expression of phospholipase A2 group IIA (*PLA2G2A*) was upregulated by oncogenic *K-ras*. The elevated expression of PLA2G2A was also observed in pancreatic cancer tissues and was correlated with poor survival of PDAC patients. Abrogation of PLA2G2A by siRNA or by pharmacological inhibition using tanshinone I significantly increased lipid peroxidation, reduced fatty acid synthase (FASN) expression, and impaired mitochondrial function manifested by a decrease in mitochondrial transmembrane potential and a reduction in ATP production, leading to the inhibition of cancer cell proliferation. Our study suggests that high expression of *PLA2G2A* induced by oncogenic *K-ras* promotes cancer cell survival, likely by reducing lipid peroxidation through its ability to facilitate the removal of polyunsaturated fatty acids from lipid membranes by enhancing the de novo fatty acid synthesis and energy metabolism to support cancer cell proliferation. As such, PLA2G2A might function as a downstream mediator of K-ras and could be a potential therapeutic target.

## 1. Introduction

The vast majority of pancreatic ductal adenocarcinoma (PDAC) is characterized by the presence of mutant *K-ras* and is one of the deadliest malignancies with a 5-year survival rate of less than 10% [1,2]. Currently, surgery followed by chemotherapy and neoadjuvant chemotherapy remain the potential effective treatments to cure this disease. However, the majority of patients are diagnosed at an unresectable stage, mainly due to the lack of biomarkers and tools for early diagnosis [3]. Targeting K-ras-activated signaling pathways has been considered as a potential strategy to develop new approaches for the treatment of PDAC.

K-ras is a small GTPase and is active when it binds guanosine triphosphate (GTP) [4]. Missense mutations in the *K-ras* gene are often found in cancers and are mainly located at one of the three mutational hotspots: glutamine-61 (Q61), glycine-12 (G12), or glycine-13 (G13). Among them, G12D mutation is the major *K-ras* mutation in PDAC. Previous studies revealed that K-ras activation by mutations persistently stimulated multiple signaling pathways and reprogrammed metabolism to drive tumor growth [1,5,6]. Although active lipid metabolism has been observed in pancreatic cancer cells and is considered an important phenotype of PDAC cells, the mechanisms by which K-ras regulates lipid metabolism await further investigation.

Phospholipase A2 (PLA2) is a type of enzyme that hydrolyzes phospholipids at the sn-2 position and releases polyunsaturated fatty acids (PUFA) [7]. The 20-carbon PUFA are metabolic precursors of bioactive eicosanoids through a series of enzymatic conversion mediated by cyclooxygenase (COX), lipoxygenase (LOX), and cytochrome P450 (CYP) [8]. These molecules, especially prostaglandins (PGs), play important roles in inflammation and in promoting a pro-tumorigenic environment [9,10,11]. Phospholipase A2 group IIA (PLA2G2A) is one of the well-studied isoforms of PLA2. Previous studies showed that mammalian PLA2G2A contributed to inflammation and host defense against certain bacteria [12,13] and seemed to be associated with poor therapeutic response and shorter survival in patients with esophageal adenocarcinoma and glioblastoma [14,15,16]. However, several studies conversely found that overexpression of PLA2G2A inhibited gastric adenocarcinoma and intestinal tumorigenesis and was associated with improved patient survival [17,18,19,20]. These studies suggest that the pro-tumorigenic or anti-tumorigenic effects of PLA2G2A appear to be tissue-specific. Currently, there have been no reports on the roles of PLA2G2A in K-ras-driven PDAC.

Specific inhibitors of PLA2G2A would be very useful to further evaluate its seemingly paradoxical roles in cancer development. There is only very limited information on pharmacological inhibitors of PLA2G2A reported in the literature. Tanshinone I is one of the major pharmacologically active compounds in Dan-shen, which is a cardioprotective herb traditionally used in Asian countries [21]. Kim et al. demonstrated that tanshinone I decreased the levels of cellular eicosanoids by inhibiting PLA2G2A activity [22]. As such, this compound, in conjunction with specific siRNAs, would provide useful tools to test the role of PLA2G2A in affecting lipid metabolism and the cellular behaviors of pancreatic cells driven by oncogenic *K-ras*.

In the present study, we first found that the mRNA expression of *PLA2G2A* was upregulated by K-ras^G12D^ activation, and high mRNA expression of *PLA2G2A* was significantly correlated with poor survival of patients with PDAC. Suppression of PLA2G2A by siRNA silencing or by pharmacological inhibition (tanshinone I) increased cellular lipid peroxidation, reduced the expression of fatty acid synthase (FASN) involved in de novo fatty acids synthesis, and impaired mitochondrial oxidation phosphorylation function, leading to the inhibition of PDAC cell proliferation. Therefore, our study revealed that targeting PLA2G2A could provide a new potential therapeutic strategy to eliminate K-ras-driven PDAC cancer cells.

## 2. Results

### 2.1. Activation of Mutant K-ras^G12D^ Induces PLA2G2A Expression Which Is Correlated with Poor Prognosis in PDAC Patients

Phospholipase PLA2G2A hydrolyzes cell membrane phospholipids and releases PUFA, which are then utilized by COX and CYP to produce eicosanoids that could promote cancer progression (Figure 1A). To explore the potential roles of PLA2G2A in PDAC, we first used a doxycycline-inducible K-ras expression in the pancreatic cell model, HPNE/*K-ras*^G12D^, to test the effect of oncogenic *K-ras* activation on the expression of PLA2G2A. We first found that *PLA2G2A* mRNA expression was significantly higher in HPNE/*K-ras*^G12D^/ON cells compared to the HPNE/*K-ras*^G12D^/OFF cells (Figure 1B), indicating that this molecule might be important for K-ras-driven PDAC cells. To further evaluate the role of PLA2G2A in PDAC, we used the pancreatic cancer datasets from SurvExpress [23] and GEPIA [24], which contained RNA sequencing (RNA-seq) data and survival information on the PDAC patients. As shown in Figure 1C, our analysis revealed that *PLA2G2A*, prostaglandin-endoperoxide synthase 1 (*PTGS1)*, and cytochrome P450 family 2 subfamily J member 2 *(CYP2J2)* expressions were positively correlated with the *K-ras* mRNA level, all with a *p* value of less than 0.001, suggesting that PLA2G2A-mediated eicosanoids synthesis might be activated by K-ras and could likely play a role in PDAC development. We then analyzed the expressions of these genes in PDAC samples in comparison with normal tissues and found that these genes were consistently upregulated in PDAC tumor tissues (Figure 1D). Further analysis showed that the expression of *PLA2G2A*, *PTGS1*, or *CYP2J2* was correlated with poor clinical outcome in terms of overall survival (Figure 1E).

### 2.2. Abrogation of PLA2G2A Suppresses PDAC Cell Proliferation

To test the impact of PLA2G2A on K-ras-driven PDAC cells, we used different experimental methods to evaluate the effect of PLA2G2A on PANC-1 and AsPC-1 pancreatic cells harboring mutant *K-ras* (G12D mutation). As shown in Figure 2A,B, a specific knockdown of *PLA2G2A* expression using siRNA significantly inhibited PANC-1 and AsPC-1 cell proliferation. Interestingly, the suppression of PLA2G2A expression resulted in a 55% decrease in the proliferation of HPNE/*K-ras^G12D^*/ON cells with stable induction of K-ras expression, whereas less inhibition (30%) was observed in HPNE/*K-ras^G12D^*/OFF cells (Figure 2B). These findings suggest that cells with K-ras activation were more sensitive to inhibition of PLA2G2A. Tanshinone I, a PLA2G2A inhibitor, was then used as a second method to test its impact on the proliferation of PDAC cell lines. MTT assay showed PANC-1 and AsPC-1 cell growth were both inhibited after tanshinone I treatment (Figure 2C). Colony formation assay also revealed that inhibition of PLA2G2A by tanshinone I (5 μM) almost completely abolished the ability of AsPC-1 and PANC-1 cells to form colonies (Figure 2D,E). Together, these data suggest that PLA2G2A plays an important role in the proliferation of PDAC cells.

### 2.3. Abrogation of PLA2G2A Induces a Moderate Increase in Lipid Peroxidation without Causing Ferroptosis

Based on the function of PLA2G2A in lipid membrane remodeling through enzymatic removal of PUFA that increase its vulnerability to lipid peroxidation (Figure 3A), we first tested whether abrogation of PLA2G2A could cause an increase in lipid peroxidation. As shown in Figure 3B,C, siRNA knockdown of *PLA2G2A* in PANC-1 and AsPC-1 cells for 72 h caused a moderate but significant increase in lipid peroxidation, as detected by an increase in BODIPY^TM^ 581/591 C11 fluorescence measured by flow cytometry analysis. Consistently, inhibition of PLA2G2A by tanshinone I also caused an accumulation of lipid peroxidation both in PANC-1 cells and in AsPC-1 cells (Figure 3D,E). Since lipid peroxidation could cause an iron-dependent cell death known as ferroptosis [25], we used ferrostatin-1, an inhibitor of ferroptosis, to test if it could affect tanshinone I-induced cell death. The results showed that ferrostatin-1 could not rescue the cells from the impact of tanshinone I, as evidenced by no changes in the tanshinone I-induced decrease of cells (Figure 3F). These data together suggest that the modest lipid peroxidation due to PLA2G2A inhibition by tanshinone I was insufficient to trigger ferroptosis, and that other mechanisms should be explored to understand the impact of PLA2G2A on cell survival and proliferation. 

### 2.4. Inhibition of PLA2G2A Significantly Suppresses the Expression of FASN Involved in De Novo Fatty Acid Synthesis

To explore the mechanisms contributing to the effect of PLA2G2A on cancer cell proliferation and survival, we tested the impact of PLA2G2A on cellular fatty acid metabolism in pancreatic cancer cells. As shown in Figure 4A, a knockdown of *PLA2G2A* by siRNA significantly suppressed the expressions of genes involved in de novo fatty acid synthesis in PANC-1 cells. Notably, FASN and stearoyl-CoA desaturase (SCD) were dramatically suppressed when *PLA2G2A* was silenced by siRNA in PANC-1 cells. A knockdown of *PLA2G2A* expression caused a significant decrease in FASN expression in AsPC-1 cells without obvious changes in the expressions of other genes involved in fatty acid synthesis (Figure 4A). Western blot analysis further showed that FASN protein expression was also decreased in the *PLA2G2A* knockdown PANC-1 and AsPC-1 cells (Figure 4B).

To further examine the relationship among K-ras, PLA2G2A, and FASN, we first used the GEPIA dataset containing RNA-seq data of pancreatic cancer from TCGA and the corresponding normal tissues from GTEx to analyze the potential correlation among these molecules in clinical samples. Spearman correlation analysis revealed that *K-ras* mRNA expression was positively correlated with the level of *FASN* mRNA, which was also correlated with the *PLA2G2A* mRNA level (Figure 4C). We then used the HPNE/*K-ras^G12D^* cell model to further test their regulatory relationship. As shown in Figure 4D,E, *PLA2G2A* and *FASN* mRNA levels were significantly suppressed when K-ras^G12D^ was turned off (without doxycycline). Western blot analysis also showed that FASN protein expression in HPNE/*K-ras^G12D^*/OFF cells was lower than that in HPNE/*K-ras^G12D^*/ON cells (Figure 4F). We also used siRNA to further test whether a knockdown of PLA2G2A could affect FASN levels in HPNE/*K-ras^G12D^*/OFF and *K-ras^G12D^*/ON cells. As shown in Figure 4G–I, *FASN* mRNA and protein levels were both decreased in HPNE/*K-ras^G12D^*/OFF and *K-ras^G12D^*/ON cells with *PLA2G2A* knockdown, suggesting that the regulation of FASN by PLA2G2A was likely conservative, and could occur regardless of *K-ras* status.

Consistent with these findings, the mRNA expressions of the genes involved in de novo fatty acid synthesis as well as FASN proteins were suppressed significantly in PANC-1 treated with 10 µM tanshinone I (Figure 5A,B). Tanshinone I at 10 µM also suppressed the mRNA expressions of these genes and the FASN protein level in AsPC-1 cells, HPNE/*K-ras^G12D^*/OFF, and *K-ras^G12D^*/ON cells (Figure 5C–H). Notably, at lower concentrations (1–5 µM), tanshinone I seemed to increase the expressions of fatty acid synthesis genes in HPNE/*K-ras^G12D^*/OFF cells (Figure 5E). These data together suggest that inhibition of de novo fatty acid synthesis by PLA2G2A abrogation could be a mechanism contributing to the suppression of cell proliferation.

### 2.5. Inhibition of PLA2G2A Causes Mitochondria Dysfunction in PDAC Cells

Mitochondria are important organelles that provide energy and metabolic intermediates for many biochemical pathways to support cell survival and proliferation [26,27]. Mitochondrial dysfunction, often associated with *K-ras* mutations, has been observed in cancers and as contributing to tumorigenesis [28,29]. We also previously showed that oncogenic *K-ras* induced optic atrophy 3 (OPA3) expression to maintain mitochondrial function and energy metabolism [30]. Based on these observations and the finding that PLA2G2A might be located on the outer mitochondria membrane [31], we further evaluated the impact of PLA2G2A on mitochondria function in pancreatic cancer cells. We found that *PLA2G2A* knockdown caused a significant loss of mitochondrial transmembrane potential in a significant portion of PANC-1 and AsPC-1 cells (Figure 6A), as detected by a decrease in rhodamine-123 fluorescent intensity in 14.9% of PANC-1 cells and 18.6% of AsPC-1 cells (Figure 6B). Further analysis of cellular ATP showed that a knockdown of *PLA2G2A* expression in human pancreatic cancer cells led to a significant decrease in cellular ATP level (Figure 6C). The NADH:ubiquinone oxidoreductase subunit B8 (NDUFB8, a subunit of mitochondrial respiratory complex I) protein expression also decreased in PANC-1 and AsPC-1 cells with knockdown of *PLA2G2A* (Figure 6D), consistent with the decrease in ATP level. Interestingly, PLA2G2A inhibition induced a modest increase in the mitochondrial reactive oxygen species (ROS) level in PANC-1 cells, but a decrease in AsPC-1 cells (Figure 6E) for a yet unknown reason. Treatment with tanshinone I led to a significant decrease in cellular ATP (Figure 6F) and a substantial increase in mitochondrial ROS in both pancreatic cancer cells (Figure 6G), suggesting that inhibition of PLA2G2A could impair mitochondrial function with a decrease in ATP production.

Taken together, these data showed that a high expression of *PLA2G2A* induced by K-ras activation could promote PDAC cell proliferation by facilitating the removal of PUFA to reduce lipid peroxidation, by upregulating FASN to promote de novo fatty acid synthesis, and by enhancing mitochondrial energy production to support cancer cell survival and growth. As such, inhibition of PLA2G2A could be a potential therapeutic strategy to suppress K-ras-driven cancer (Figure 7).

## 3. Discussion

Although many studies have shown that oncogenic *K-ras* mutation could activate multiple signaling pathways, the roles and mechanisms by which *K-ras* regulates lipid membrane remodeling in PDAC cells still remain unclear. Phospholipase PLA2G2A hydrolyzes phospholipids at the sn-2 position and releases PUFA. Proper turnover of PUFA in lipid membranes is important for the maintenance of membrane integrity since abnormal accumulation of PUFA is vulnerable to lipid peroxidation. However, there is no report in the literature on the role of PLA2G2A in K-ras-driven PDAC cells.

In this study, we first found that expression of *PLA2G2A* was upregulated after K-ras^G12D^ activation. Through analysis of the public database, we also found that the expression of *PLA2G2A* in PDAC tissues was higher than that in normal tissues, and the high *PLA2G2A* expression was associated with poor clinical outcomes of PDAC patients. Thus, the upregulation of *PLA2G2A* by K-ras observed in the cell model is likely relevant to the findings in clinical samples. However, the regulatory mechanisms of PLA2G2A expression in K-ras-driven cancer cells remain unknown. A potential molecular pathway could be that K-ras activates the NF-κB signaling pathway [32], which in turn induces *PLA2G2A* expression [33].

A significant new finding from this study was that PLA2G2A could promote PDAC proliferation via enhancing the de novo synthesis of fatty acids. We observed that knockdown of *PLA2G2A* consistently led to a significant downregulation of FASN expression in all cell lines tested (HPNE/*K-ras^G12D^*/ON cells, *K-ras^G12D^* /OFF cells, PANC-1, and AsPC-1 cells), while the expressions of SCD and other lipid metabolic enzymes also showed various degrees of downregulation but with a notable exception in AsPC-1 cells. Based on these observations, we postulate that FASN might be the key PLA2G2A-regulated molecule that promotes de novo fatty acid synthesis. It should be noted that although K-ras could promote lipid metabolism through upregulation of PLA2G2A expression, the regulation of FASN expression by PLA2G2A, per se, appeared independent of K-ras expression. Thus, PLA2G2A seems to be a downstream effector regulated by oncogenic *K-ras*, while FASN expression is likely PLA2G2A-dependent. As such, PLA2G2A could regulate FASN expression in HPNE/*K-ras^G12D^*/OFF cells (Figure 4G,I). The regulation of FASN expression by PLA2G2A seems conservative in all four cell lines tested, but the underlying mechanisms still remain unclear and require further study. Another novel finding from this study was the discovery that PLA2G2A plays a significant role in promoting mitochondrial energy metabolism, as evidenced by the impairment of mitochondrial transmembrane potential and the reduction in ATP production. Previous studies in our lab have demonstrated that oncogenic *K-ras* activation led to mitochondrial dysfunction and high generation of ROS [29], and OPA3 overexpression induced by K-ras could promote mitochondrial energy metabolism as a mechanism to offset the negative impact of K-ras on mitochondria [30]. The present study suggests that K-ras-induced *PLA2G2A* expression might be another mechanism to promote mitochondrial ATP generation and maintain energy homeostasis. However, the mechanistic links between PLA2G2A and mitochondrial energy metabolism remain unclear. One possibility is that the enzymatic activity of PLA2G2A facilitates the removal of PUFA from lipid membranes to reduce their vulnerability to lipid peroxidation and thus helps to maintain mitochondrial membrane integrity, while the released PUFA could be used as the fuel for ATP generation.

The important role of PLA2G2A in promoting cell survival and proliferation in K-ras-driven cancer cells suggests that it could be a potential therapeutic target. Although there is currently no specific inhibitor of PLA2G2A for the clinical treatment of cancer, tanshinone I has been reported to inhibit PLA2G2A and could disrupt cellular PUFA metabolism for the synthesis of eicosanoids [23]. This compound has been shown to exert antitumor effects by inducing apoptosis through the generation of reactive oxygen species in human hepatocellular carcinoma [34] and endometrial cancer cells [35], and has been shown to inhibit cell proliferation by blocking the cell cycle and autophagy in various tumor cell models [36]. Recently, tanshinone I was also proposed as an inhibitor of the enhancer of zeste homolog 2 (EZH2) [37], suggesting that tanshinone I might be a multi-targeting compound, and its biological effect is likely the combined results of its actions on multiple targets. Nevertheless, we found that tanshinone I could cause a series of changes highly similar to those induced by siRNA knockdown of *PLA2G2A* in pancreatic cancer cells, suggesting that inhibition of PLA2G2A is likely a major mechanism of tanshinone I. Since tanshinone I is an active component of Dan-shen, which is used in tradition Chinese medicine for treatment of certain illnesses safely, it would be feasible to use tanshinone I for potential treatment of K-ras-driven cancer. However, it should be noted that the current study mainly used cell lines for in vitro experiments, which only demonstrated the impact of PLA2G2A abrogation on the biochemical and molecular events in cell culture and might not be able to accurately predict what would happen in vivo. Thus, further experiments using organ-on-a-chip models that mimic in vivo settings and animal studies are needed to further validate the therapeutic relevance of targeting PLA2G2A activity for cancer treatment. Since our study showed that the inhibition of PLA2G2A exhibited some degree of growth inhibition in HPNE/*K-ras^G12D^*/OFF cells, toxicity to normal cells should be a potential concern when considering the use of tanshinone I in vivo. Interestingly, an in vivo study showed that tanshinone I at a dosage of up to 200 mg/kg in mice did not cause apparent toxicity in the animals, suggesting that this compound could be well-tolerated in vivo without severe toxic side effects [38,39]. Obviously, the drug safety and therapeutic efficacy still need to be further evaluated in clinical trials.

In summary, our study showed that oncogenic *K-ras* activation could upregulate the expression of *PLA2G2A*, which in turn promotes PDAC cell survival and proliferation by facilitating the removal of PUFA to reduce cellular lipid peroxidation, promoting FASN expression involved in de novo fatty acid synthesis and mitochondrial ATP production. As such, PLA2G2A could serve as a potential therapeutic target. Although tanshinone I might have multiple targets in the cells, its ability to inhibit PLA2G2A and suppress K-ras-driven cancer proliferation provides a possibility for use in treatment of cancers with *K-ras* mutations.

## 4. Materials and Methods

### 4.1. Cell Lines

Human pancreatic PANC-1 (Cat#CRL-1469) and AsPC-1 (Cat#CRL-1682) cancer cell lines, both harboring *K-ras^G12D^* mutation, were purchased from ATCC (Manassas, VA, USA). PANC-1 cells were maintained in DMEM supplemented with 10% fetal bovine serum (FBS). AsPC-1 cells were maintained in RPMI-1640 supplemented with 10% FBS. The doxycycline-inducible HPNE/*K-ras^G12D^* cells were generated as previously described [40] and cultured in 35% DMEM, 35% F-12K medium, and 20% M3 base medium supplemented with 10% tetracycline-free FBS. All cell lines were tested for mycoplasma using a mycoplasma PCR detection kit (Sigma, Saint-Louis, MO, USA) and maintained in a humidified incubator with 5% CO_2_ at 37 °C.

### 4.2. Cell Culture Reagents

Tanshinone I (MedChemExpress, Cat#HY-N0134) and Ferrostatin-1 (Selleck, Cat# S7243) were dissolved in dimethyl sulfoxide (DMSO) at 10 mM stock concentration and stored in aliquots at −20 °C. Thiazolyl blue tetrazolium bromide (MTT, Sigma, Cat#M2128) was dissolved in phosphate-buffered saline (PBS) at 5 mg/mL concentration and stored in aliquots at 4 °C. Crystal violet staining solution (Cat#C0121) was purchased from Beyotime Biotechnology.

### 4.3. Bioinformatics

The relationship between genes involved in eicosanoids synthesis (*PLA2G2A, PTGS1, CYP2J2*) and 189 PDAC patients’ prognoses were determined using SurvExpress database (http://bioinformatica.mty.itesm.mx/SurvExpress, accessed on 2 November 2021). The correlation between the above genes and *K-ras* expression, and the mRNA levels of these genes in tumor and normal pancreatic tissues were both measured using the online database Gene Expression Profiling Interactive Analysis (GEPIA, http://gepia.cancer-pku.cn, accessed on 14 March 2022).

### 4.4. Cell Transfection

Transfection was performed on cells at 25% confluence for 72 h with a 50 nM siRNA solution using Lipofectamine RNAi Max (ThermoFischer, Rockford, IL, USA) according to the manufacturer’s protocol. Cells were then washed with PBS and replaced with DMEM for further experiments. The small interfering RNAs (siRNAs) against human *PLA2G2A* were synthesized by RiboBio (Guangzhou, China). The sequence of siRNA for human *PLA2G2A* is: 5′- GAAACAAGACGACCTACAA-3′.

### 4.5. MTT Assay

Cell viability and proliferation were measured using MTT assay. Briefly, cells were seeded in a 96-well plate and treated with the indicated doses of tanshinone I for 3, 5, and 7 days (1 × 10^3^ cells per well, 200 μL culture medium per well). Then, MTT solution was added into each well (final concentration: 0.5 mg/mL) and cells were incubated for 4 h at 37 °C. The culture medium was then discarded, and the blue formazan crystals were solubilized with DMSO (200 µL/well). The colored solution was subsequently read at 570 nm.

### 4.6. Colony Formation Assay

Cells were cultured in 6-well plates (500 cells per well) and treated with 5 μM tanshinone I. After 2 weeks of incubation, cells were fixed with methanol solution for 30 min, stained with crystal violet staining solution for 30 min, washed with water, and finally air dried. Data were expressed by the number of colonies.

### 4.7. Quantitative Reverse Transcription-Polymerase Chain Reaction (qRT-PCR)

Total RNA was isolated using the RNA-Quick Purification Kit (ESscience, Shanghai, China) according to the manufacturer’s instructions. RNA was reverse-transcribed using the Primer Script RT reagent Kit with gDNA Eraser (Takara BIO INC, Japan). PCR was performed using the SYBR Premix Ex Taq RNAse H+ kit (Takara BIO INC, Japan) and analyzed using the Bio-Rad detection system (Bio-Rad, Hercules, CA, USA). The samples were first incubated 5 min at 95 °C, followed by 40 cycles of 10 s at 95 °C and 30 s at 60 °C. The results were calculated (formula: 2^-(Ct target-Ct actin)^) and matched to the control samples. The primers sequences are: human *PLA2G2A* (F:5′-GAAAGGAAGCCGCACTCAGTT-3′, R:5′-CAGACGTTTGTAGCAACAGTCA-3′); human *SREBF1* (F:5′-ACAGTGACTTCCCTGGCCTAT-3′, R:5′-GCATGGACGGGTACATCTTCAA-3′); human *ACACA* (F:5′-TCACACCTGAAGACCTTAAAGCC-3′, R:5′-AGCCCACACTGCTTGTACTG-3′); human *ACACB* (F:5′-AGAAGACAAGAAGCAGGCAAAC-3′, R:5′-GTAGACTCACGAGATGAGCCA-3′); human *ACLY* (F:5′-ATCGGTTCAAGTATGCTCGGG-3′, R:5′-GACCAAGTTTTCCACGACGTT-3′); human *ACSS2* (F:5′-AAAGGAGCAACTACCAACATCTG-3′, R:5′-GCTGAACTGACACACTTGGAC-3′); human *FASN* (F:5′-ACAGCGGGGAATGGGTACT-3′, R:5′-GACTGGTACAACGAGCGGAT-3′); human *SCD* (F:5′-TTCCTACCTGCAAGTTCTACACC-3′, R:5′-CCGAGCTTTGTAAGAGCGGT-3′); human *β-actin* (F:5′-AACTCCATCATGAAGTGTGAC-3′, R:5′-GATCCACATCTGCTGGAAGG-3′).

### 4.8. Western Blotting

Cells were washed twice with ice-cold PBS at the indicated time and lysed in lysing buffer (5 g SDS, 0.37 g EDTA, 0.2922 g NaCl, 1 mL 1M Tris-Cl, adjust pH to 7.5 with HCl and add ddH_2_O to 100 mL). The concentration of proteins was quantified using a BCA (bicinchoninic acid) protein assay (Thermo Fisher, Rockford, IL, USA). Protein samples were run on a standard SDS-PAGE and transferred to PVDF membranes, which were then saturated with 5% non-fat milk. Subsequently, the membranes were blotted with specific primary antibodies overnight at 4 °C. The primary antibodies used in this study were: anti-ATP5A (Cat#ab14748), anti-NDUFB8 (Cat#ab110242), and anti-FASN (Cat#ab128870) antibodies were purchased from Abcam (Cambridge, MA, USA). Anti-Vinculin (Cat#13901) antibody was purchased from Cell Signaling Technology (Cambridge, MA, USA). Then, the membranes were incubated with appropriate horseradish peroxidase-conjugated secondary antibodies, and the bands were visualized by chemiluminescence using an imaging system (Bio-Rad Laboratories, Carlsbad, CA, USA).

### 4.9. Measurement of Cellular Mitochondrial ROS, Lipid Peroxidation, and Mitochondrial Transmembrane Potential

Cells were seeded in 6-well plates (2 × 10^5^ cells per well) overnight and treated with siRNA or tanshinone I (5 µM) for the indicated times. Then, cells were incubated respectively with mitoSOX (5 µM), BODIPY 581/591 C11 (5 µM) or Rhodamine 123 (1 μM) probes at 37 °C for 30 min in order to detect superoxide anion radicals in mitochondria, lipid peroxidation, and mitochondrial transmembrane potential, respectively. Cells were then washed twice with PBS, and the corresponding ROS levels were measured using a Beckman flow cytometer (Beckman Coulter, Miami, FL, USA). BODIPY^TM^ 581/591 C11 probe (Cat#D3861) was purchased from Thermo Fisher. MitoSOX^TM^ red probe (Cat#M36008) was purchased from Invitrogen. Rhodamine 123 probe (Cat#R8004) was purchased from Sigma-Aldrich.

### 4.10. Cellular ATP Detection

To measure ATP levels, cells (5 × 10^3^) were seeded in 96-well black plates and ATP content was assessed using the bioluminescence CellTiter-Glo viability assay kit (Promega, Madison, WI, USA) according to the procedure provided by the manufacturer.

### 4.11. Statistical Analysis

All statistical analyses were performed using GraphPad Prism 7 software. A two-tail unpaired *t*-test was used for comparing two experimental groups, and one-way or two-way ANOVA was applied when comparing three or more experimental groups. Log-rank (Mantel-Cox) test was used for statistical analysis of cancer patients’ survival. Despite a large sample size, the relationship between K-ras and eicosanoids-synthesis-related proteins expression in human pancreatic carcinoma tissues was assessed using a Spearman’s rank correlation because of the nature of the data (integer scores). A *p*-value of 0.05 or less was considered as statistically significant and was indicated as follows: * *p* < 0.05; ** *p* < 0.01; *** *p* < 0.001; **** *p* < 0.0001.

## Figures and Tables

**Figure 1 ijms-23-11721-f001:**
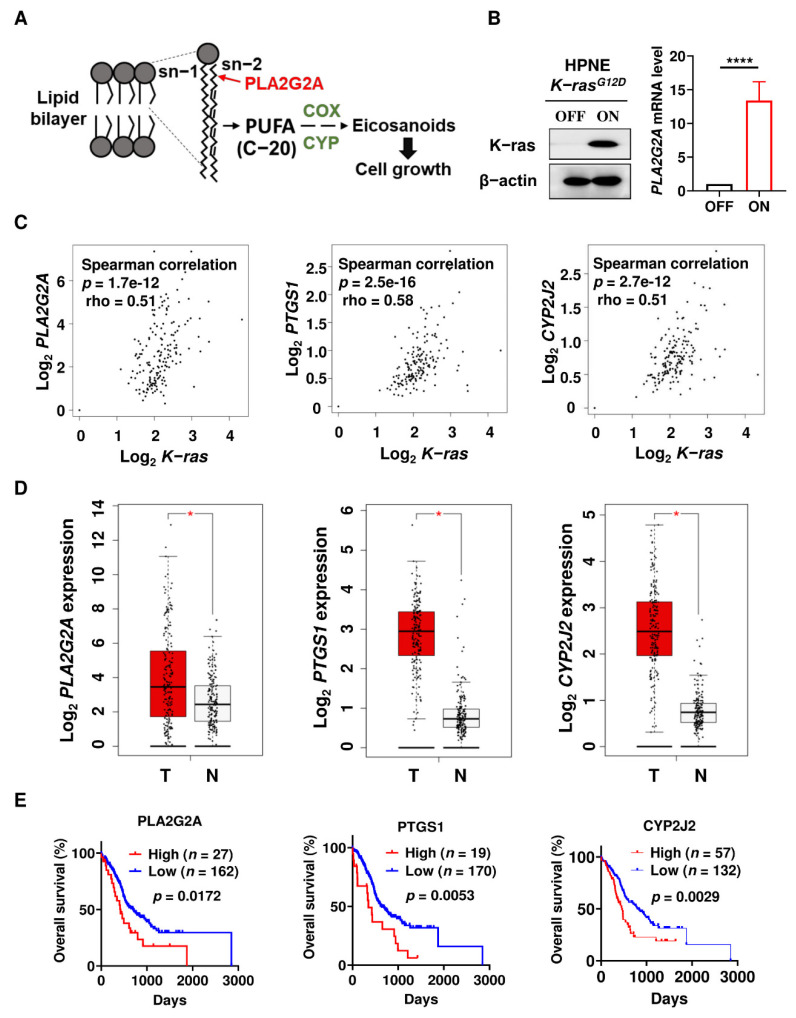
Activation of mutant *K-ras*^G12D^ induces *PLA2G2A* expression and the relationship between *PLA2G2A* expression and clinical outcomes of PDAC patients. (**A**) Schematic illustration of the PLA2G2A-mediated pathway with a release of PUFA for synthesis of eicosanoids. PLA2G2A, phospholipase A2 group IIA; PUFA, polyunsaturated fatty acid; COX, cyclooxygenase; CYP, cytochrome P450. (**B**) HPNE/*K-ras*^G12D^/OFF cells were incubated with 100 ng/mL doxycycline to induce K-ras^G12D^ expression (left panel). After 1 month of stable K-ras induction, *PLA2G2A* mRNA level was quantified by qRT-PCR (right panel). (**C**) Correlation between expressions of *K-ras* and genes involved in eicosanoids synthesis using GEPIA database. PTGS1, prostaglandin-endoperoxide synthase 1; CYP2J2, cytochrome P450 family 2 subfamily J member 2. (**D**) Comparison of mRNA expression of eicosanoid synthesis-related genes in PDAC tumor (T) tissue (*n* = 179) and normal (N) pancreatic tissues (*n* = 171) using the GEPIA datasets. (**E**) Association between PDAC patient survival and the expressions of *PLA2G2A*, *PTGS1*, and *CYP2J2*. Statistical analyses: Spearman’s rank correlation test for (**C**); unpaired *t*-test for (**D**); Log-rank test for (**E**). *, *p* < 0.05; ****, *p* < 0.0001.

**Figure 2 ijms-23-11721-f002:**
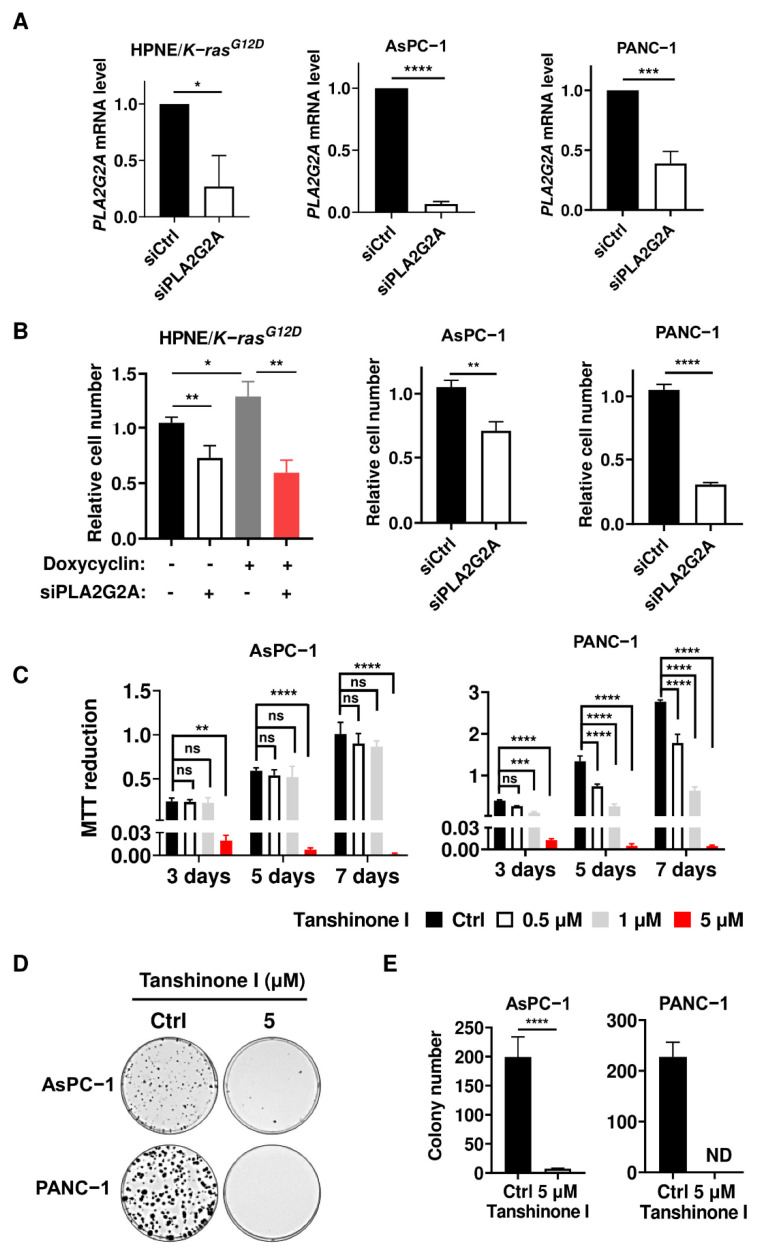
Suppression of PLA2G2A inhibits PDAC cell proliferation. (**A**) PANC-1, AsPC-1, and HPNE/*K-ras^G12D^* cells (stably induced with doxycycline to express oncogenic K-ras for 1 month) were transfected with 50 nM siRNA against *PLA2G2A* for 72 h, and the *PLA2G2A* mRNA level was quantified by qRT-PCR. (**B**) PANC-1, AsPC-1, HPNE/*K-ras^G12D^*/OFF, and *K-ras^G12D^*/ON cells were transfected with 50 nM siRNA against *PLA2G2A* for 72 h and then seeded in 6-well plates for 72 h. Cell proliferation was measured by counting the number of cells. (**C**) PANC-1 and AsPC-1 cells were treated with various concentrations (0.5–5 µM) of tanshinone I for various time periods, as indicated. Cell proliferation was analyzed by MTT assay. (**D**,**E**) PANC-1 and AsPC-1 cells were treated with 5 µM tanshinone I and incubated for two weeks to form colonies. Representative images (**D**) and quantification of colonies (**E**) are shown. Statistical analyses: data are presented as the mean ± SD of three independent experiments; unpaired *t*-test for (**A**,**B**,**E**); two-way ANOVA for (**C**). * *p* < 0.05; ** *p* < 0.01; *** *p* < 0.001; **** *p* < 0.0001; ns = non significant.

**Figure 3 ijms-23-11721-f003:**
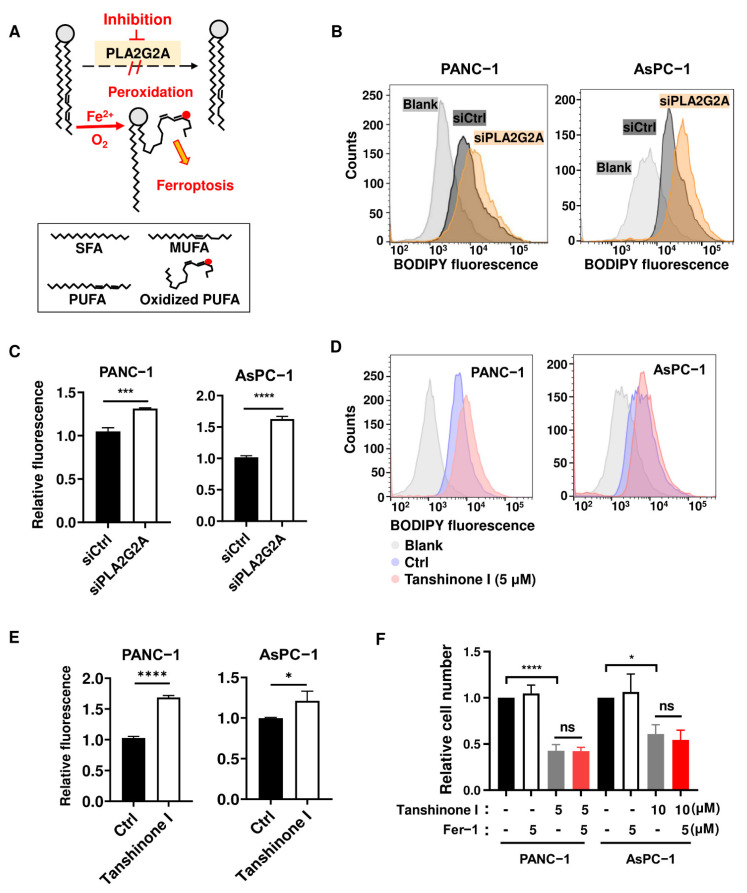
Suppression of PLA2G2A induces a moderate lipid peroxidation without inducing ferroptosis. (**A**) Schematic illustration for the role of PLA2G2A in lipid remodeling and peroxidation. Abbreviations: SFA, saturated fatty acid; MUFA, monounsaturated fatty acid; PUFA, polyunsaturated fatty acid. (**B**,**C**) PANC-1 and AsPC-1 cells were transfected with 50 nM siRNA against *PLA2G2A* for 72 h, and cells were incubated with 5 µM BODIPY™ 581/591 C11 dye for lipid peroxidation, which was analyzed by flow cytometry. Representative images (**B**) and quantitative data (**C**) are shown. (**D**,**E**) PANC-1 and AsPC-1 cells were incubated with 5 µM tanshinone I for 72 h; lipid peroxidation was analyzed by flow cytometry. Representative images (**D**) and quantitative data (**E**) are shown. (**F**) PANC-1 and AsPC-1 cells were treated with tanshinone I with or without ferrostatin-1 as indicated, and cell proliferation was measured by directly counting cell numbers. Statistical analyses: data are presented as the mean ± standard deviation (SD) of three separate experiments; unpaired *t*-test for (**C**,**E**); one-way ANOVA for (**F**). * *p* < 0.05; *** *p* < 0.001; **** *p* < 0.0001; ns = non significant.

**Figure 4 ijms-23-11721-f004:**
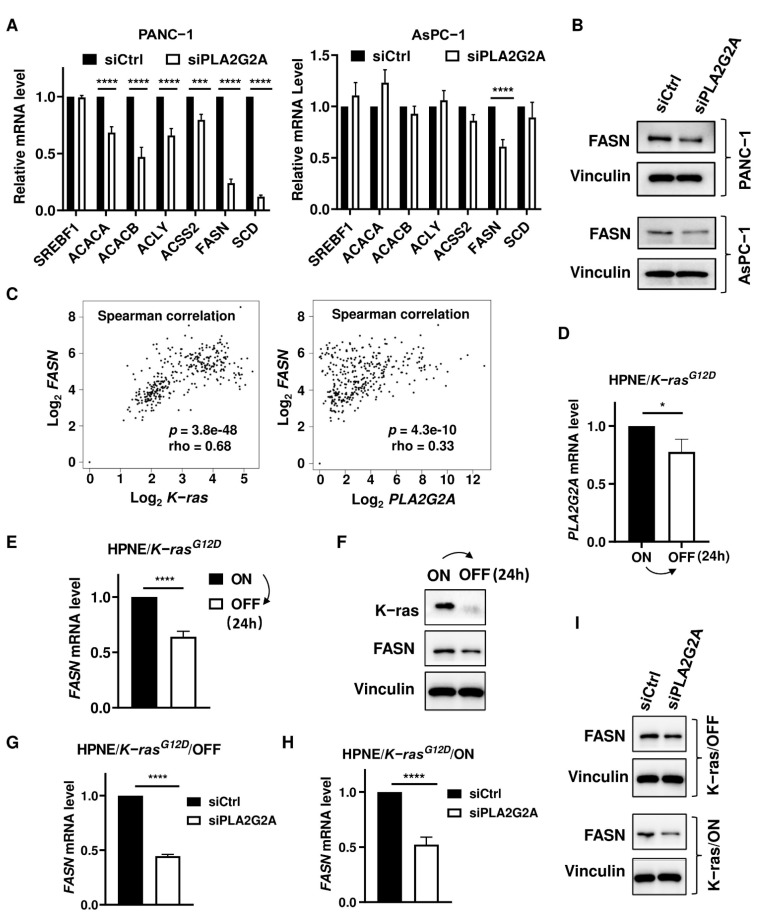
Effect of PLA2G2A on the expression of molecules involved in de novo fatty acid synthesis. (**A**) PANC-1 and AsPC-1 cells were transfected with 50 nM siRNA against *PLA2G2A* for 72 h. The mRNA levels of fatty acid synthesis-related genes were quantified by qRT-PCR. SREBF1, sterol regulatory element binding transcription factor 1; ACACA, acetyl-CoA carboxylase alpha; ACACB, acetyl-CoA carboxylase beta; ACLY, ATP citrate lyase; ACSS2, acyl-CoA synthetase short chain family member 2; FASN, fatty acid synthase; SCD, stearoyl-CoA desaturase. (**B**) PANC-1 and AsPC-1 cells were transfected with 50 nM siRNA against *PLA2G2A* for 72 h. FASN protein level was analyzed by western blot analysis (representative of 3 independent experiments). (**C**) Correlation between *K-ras* and *FASN* expression, and between *PLA2G2A* and *FASN* expression using GEPIA database. (**D**) HPNE/*K-ras^G12D^*/ON cells stably induced by doxycycline to express K-ras for 1 month were placed back into doxycycline-free medium to turn off K-ras expression for 24 h, and the mRNA level of *PLA2G2A* was quantified by qRT-PCR. (**E**) HPNE/*K-ras^G12D^*/ON cells were placed back into doxycycline-free medium to turn off K-ras expression for 24 h, and the mRNA level of *FASN* was quantified by qRT-PCR. (**F**) HPNE/*K-ras^G12D^*/ON cells were placed back into doxycycline-free medium to turn off K-ras expression for 24 h, and the protein expressions of K-ras and FASN were quantified by western blotting. (**G**) HPNE/*K-ras^G12D^*/OFF cells were transfected with 50 nM siRNA against *PLA2G2A* for 72 h, and *FASN* mRNA level was quantified by qRT-PCR. (**H**) Stable HPNE/*K-ras^G12D^*/ON cells (with doxycycline to express K-ras for 1 month) were transfected with 50 nM siRNA against *PLA2G2A* for 72 h, and *FASN* mRNA level was quantified by qRT-PCR. (**I**) HPNE/*K-ras^G12D^*/OFF and *K-ras^G12D^*/ON cells were transfected with 50 nM siRNA against *PLA2G2A* for 72 h. FASN protein level was analyzed by western blotting (representative of 3 independent experiments). Statistical analyses: data are presented as the mean ± standard deviation (SD) of three separate experiments; unpaired *t*-test for (**D**,**E**,**G**,**H**); two-way ANOVA for (**A**). * *p* < 0.05; *** *p* < 0.001; **** *p* < 0.0001.

**Figure 5 ijms-23-11721-f005:**
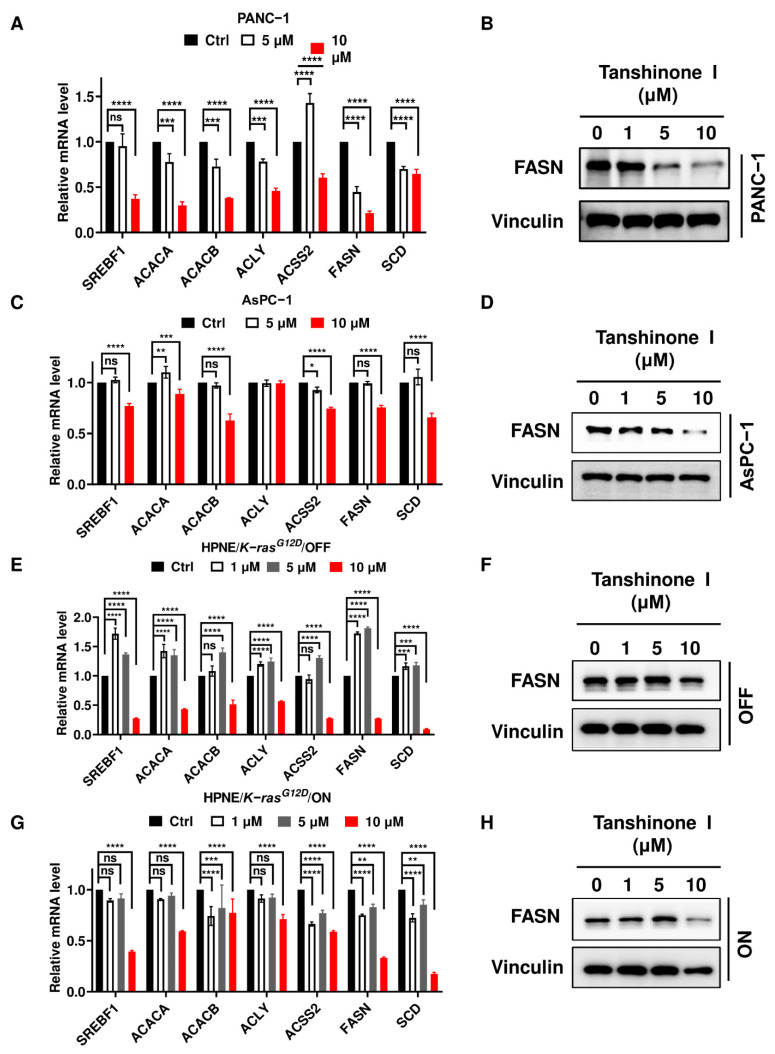
Tanshinone I treatment reduces expression of FASN involved in de novo fatty acid synthesis. (**A**) PANC-1 cells were incubated with tanshinone I (5–10 µM) for 72 h. The mRNA levels of fatty acid synthesis-related genes were quantified by qRT-PCR. (**B**) PANC-1 cells were incubated with tanshinone I (1–10 µM) for 72 h. FASN protein level was analyzed by western blot analysis (representative of 3 independent experiments). (**C**) AsPC-1 cells were incubated with tanshinone I (5–10 µM) for 72 h. The mRNA levels of fatty acid synthesis-related genes were quantified by qRT-PCR. (**D**) AsPC-1 cells were incubated with tanshinone I (1–10 µM) for 72 h. FASN protein level was analyzed by western blot analysis (representative of 3 independent experiments). (**E**) HPNE/*K-ras^G12D^*/OFF cells were incubated with tanshinone I (1–10 µM) for 72 h. The mRNA levels of fatty acid synthesis-related genes were quantified by qRT-PCR. (**F**) HPNE/*K-ras^G12D^*/OFF cells were incubated with tanshinone I (1–10 µM) for 72 h. FASN protein level was analyzed by western blot analysis (representative of 3 independent experiments). (**G**) HPNE/*K-ras^G12D^*/ON cells (induced with doxycycline to stably express K-ras for 1 month) were incubated with tanshinone I (1–10 µM) for 72 h. The mRNA levels of fatty acid synthesis-related genes were quantified by qRT-PCR. (**H**) HPNE/*K-ras^G12D^*/ON cells were incubated with tanshinone I (1–10 µM) for 72 h. FASN protein level was analyzed by western blot analysis (representative of 3 independent experiments). Statistical analyses: data are the mean ± standard deviation (SD) of three separate experiments; two-way ANOVA for (**A**, **C**, **E**, **G**). * *p* < 0.05; ** *p* < 0.01; *** *p* < 0.001; **** *p* < 0.0001; ns = non significant.

**Figure 6 ijms-23-11721-f006:**
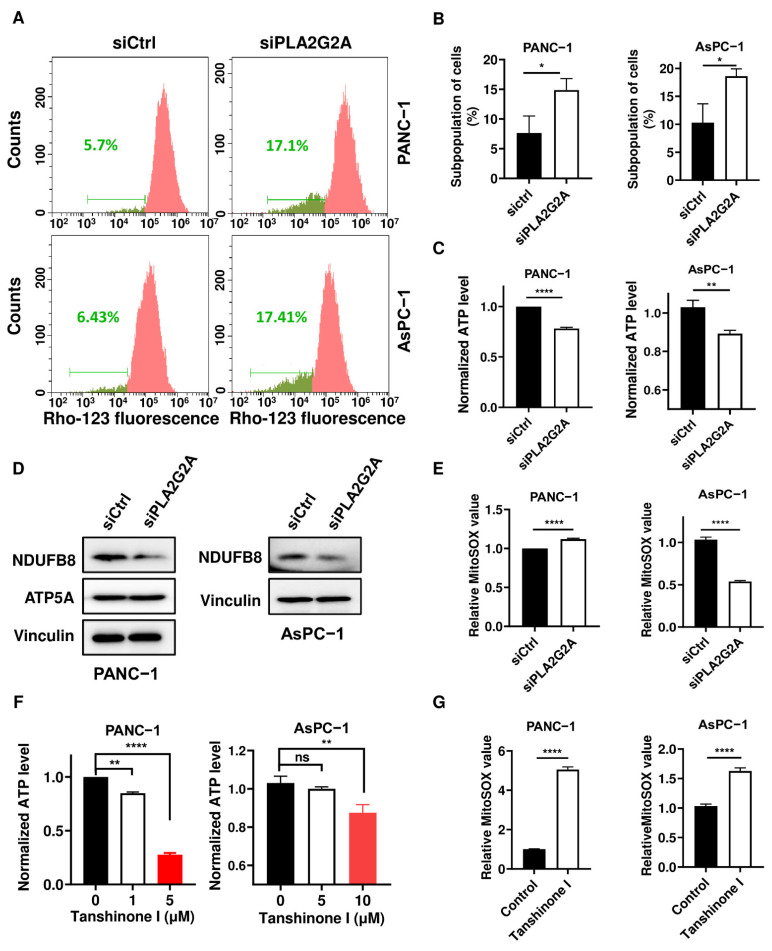
Inhibition of PLA2G2A induces mitochondrial dysfunction in PDAC cells. (**A**,**B**) Effect of *PLA2G2A* knockdown by siRNA on mitochondrial transmembrane potential. PANC-1 and AsPC-1 cells were transfected with 50 nM siRNA against *PLA2G2A* for 72 h and then stained with potential-sensitive chemical probe Rhodamine-123 followed by flow cytometry analysis. The green numbers indicate the subpopulation of cells that lost transmembrane potential in the respective samples. (**C**) Quantitation of cellular ATP in PANC-1 and AsPC-1 cells transfected with 50 nM siRNA against *PLA2G2A* or with control RNA for 72 h. (**D**) Western blot analysis of mitochondrial respiratory chain proteins in PANC-1 and AsPC-1 cells transfected with 50 nM siRNA against *PLA2G2A* or with control RNA for 72 h (representative of 3 independent experiments). NDUFB8, NADH:ubiquinone oxidoreductase subunit B8. (**E**) PANC-1 and AsPC-1 cells were transfected with 50 nM siRNA against *PLA2G2A* for 72 h. The cells were then incubated with 5 µM of MitoSOX probe for 30 min. Quantification of mitochondrial superoxide was analyzed by flow cytometry. (**F**) Quantitation of cellular ATP in PANC-1 and AsPC-1 cells incubated with the indicated concentrations of tanshinone I. (**G**) PANC-1 and AsPC-1 cells were incubated with 5 µM of tanshinone I for 72 h. Quantification of mitochondrial superoxide was analyzed by flow cytometry after the cells were stained with 5 µM of MitoSOX probe for 30 min. Statistical analyses: data are presented as the mean ± SD of three independent experiments; unpaired *t*-test for (**B**,**C**,**E**,**G**); one-way ANOVA for (**F**). * *p* < 0.05; ** *p* < 0.01; **** *p* < 0.0001; ns = non significant.

**Figure 7 ijms-23-11721-f007:**
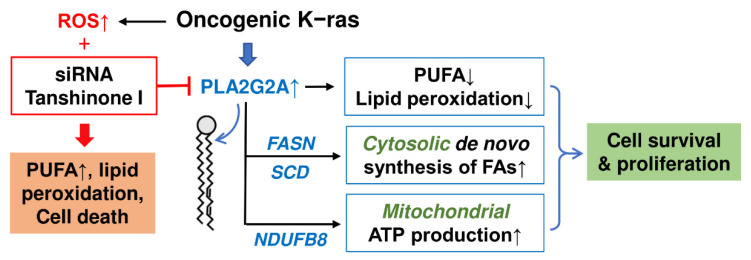
Schematic illustration of the mechanisms by which the upregulation of PLA2G2A by oncogenic *K-ras* promotes PDAC cell survival. Oncogenic *K-ras* is known to cause high ROS stress and render cancer cells more prone to ROS-induced lipid peroxidation and cell death. The elevated PLA2G2A induced by K-ras might be a novel mechanism to prevent lipid peroxidation through its ability to remove polyunsaturated fatty acids (PUFA) from lipid membranes. Since PUFA are highly vulnerable to ROS-mediated peroxidation, their removal from lipid membranes by PLA2G2A would prevent abnormal accumulation of PUFA and thus reduce the risk of lipid peroxidation and related damage. This also helps the maintenance of mitochondrial membrane integrity and metabolic functions. Furthermore, PLA2G2A upregulates the expression of FASN and other molecules involved in de novo fatty acid synthesis and promotes mitochondrial function and ATP generation. Altogether, the upregulation of PLA2G2A contributes to cell survival and proliferation of K-ras-driven cancer cells, and inhibition of PLA2G2A could abrogate these cell survival mechanisms and thus could be a potential therapeutic strategy to suppress cancer cells harboring oncogenic *K-ras*.

## Data Availability

The datasets generated or analyzed for the current study are available from the corresponding author on reasonable request.

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
