# Peer review of "PLA2G2A Phospholipase Promotes Fatty Acid Synthesis and Energy Metabolism in Pancreatic Cancer Cells with K-ras Mutation"

_ijms, 2022, doi:10.3390/ijms231911721_

Round 1
Reviewer 1 Report
Major comments:
1. It has already been shown by Yang et al, 2013 that Tanshione I has anti-cancer effects.
2. The image quality needs improving. All of the figures need improvement.
3. Why were AsPc-1 and PANC-1 selected for the study? What is the importance of these two cell lines specifically for PLA2G2A signaling?
4. The study lacks gain of-function, loss of function and recovery of function experiments to confirm the hypothesis.
5. Several experiments in the study are only performed in only one cell line (Fig. 3, 4, 5). One cell line is not adequate to test or confirm a hypothesis. The authors need to consistently confirm the experiments in multiple cell lines.
6. Figure 6, Tanshione I is mispelled.
7. The experiments should be confirmed in an organ-on-a-chip model for biological relevancy.
Reviewer 2 Report
In this study, the authors elucidated that oncogenic Kras activation upregulated PLA2G2A expression, which promoted PDAC cell survival and proliferation by using doxycycline-inducible oncogenic Kras expression and siRNA of PLA2G2A as well as PLA2G2A inhibitor. They suggested the underlying mechanisms: inhibiting lipid peroxidation, de novo fatty acid synthesis, and maintaining mitochondrial function. In addition, by using public database, they also showed that the association of oncogenic Kras expression and PLA2G2A, PTGS1 and CYP2J2, and high expression of those was associated with poor prognosis in PDAC patients.
This manuscript tried to provide some mechanistic insights of lipid metabolism, but it seems not sufficient. I raised several points to be addressed.
1) All the experiments here were in vitro ones, which is one of the limitations. Since in vitro proliferation was shown by inhibiting PLA2G2A, in vivo tumor growth with or without inhibiting PLA2G2A might strengthen this study.
2) HPNA/KrasG12D cells clearly showed the difference of PLA2G2A expression between Kras ON and OFF. The authors had better show the difference of proliferation between the ON and OFF, and also need to show the difference with or without PLA2G2A inhibitor especially in the Kras-ON cells.
3) Difference in the proliferation and colony-formation in Figure 2 is dramatic. However, I’m afraid if this inhibition might cause severe adverse effect on normal cells. If the effect of PLA2G2A inhibition is dependent on oncogenic Kras, it might be good. So, the authors had better check the proliferation with or without PLA2G2A inhibition in Kras-OFF cells or certain normal cells.
4) Most of the mechanistic experiments were done in PANC-1 cells. If these experiments are also done using AsPC-1 or other Kras-mutant PDAC cells and similar results are obtained, it might strengthen this study a lot.
5) The proliferation of PANC-1 cells was significantly inhibited at 1 microM of Tanshinone I in Figure 2C, however, most of other experiments in Figure 3-5 were done at higher concentration, especially in Figure 4D. To explore the underlying mechanism of proliferation inhibition, these experiments had better be done at 1 microM. In related to the comment of adverse effect, it is unclear if the higher concentration is appropriate for the treatment and also for these experiments.
Minor points
6) In Introduction line 31, currently neoadjuvant chemotherapy is broadly performed in clinical practice.
7) In Figure 2C (in line 113-4), the AsPC1 growth can’t be described as inhibited by tanshinone I in a dose-dependent and time-dependent manner.
8) In Figure 2E, statistical significance should be shown in PANC-1 cells.
9) In Figure 3F, statistical significance should be shown between the tanshinone I (-)+Fer-1(-) and the tanshinone I (+)+Fer-1(-), not between the tanshinone I (-)+Fer-1(+) and the tanshinone I (+)+Fer-1(-).
10) In Methods section, description about flow cytometry analysis didn’t contain rhodamine-123. The title of 4.10 should contain lipid peroxidation and mitochondrial transmembrane potential, not only cellular ROS.
Round 2
Reviewer 1 Report
1. It has already been shown by Yang et al, 2013 that Tanshione I has anti-cancer effects.
Response: We thank the reviewer for this comment, and have carefully searched the literature and found several publications by Yang et al on Tanshinone IIa, but unable to find the specific publication by Yang et al (2013) on the anticancer activity of Tanshinone I. Nevertheless, we are aware of the previous publications on antitumor activity of Tanshinone I and have already cited some of these publications in the discussion section (references 35-36; lines 275-277).
Reviewer: Could the authors please point out the originality of the work in this context? Tanshione I has been shown to have anti tumor effects and fatty acid metabolism (line 273-274). What is the novelty in its connection with K-ras and PLA2G2A and lipid peroxidation/synthesis? A schematic diagram clarifying this novelty in mechanistics would be helpful.
2. The image quality needs improving. All of the figures need improvement.
Response: As suggested, we have now improved the quality of all figures by
increasing the image resolution.
Reviewer: Image 1 is quite fuzzy, and other images are also not very clear.
3. Why were AsPc-1 and PANC-1 selected for the study? What is the importance of these two cell lines specifically for PLA2G2A signaling?
Response: This study aimed to investigate alterations in lipid metabolism in K-ras-driven cancers. Both PANC-1 and AsPC-1 cell lines are human pancreatic cancer cells widely utilized in pancreatic cancer research, and they harbor K-ras G12D mutation (lines 112-113; line 304-305).
Reviewer: PLA2G2A is upregulated by K-ras, so it would be helpful to have K-ras negative cell lines as a negative control/ loss of function studies. The K-ras negative cell line could have K-ras induced and then studied for gain of function/recovery of function as well. This could potentially be done with conditional induction of K-ras in HPNE/KrasG12D or knocking out K-ras or using the parental HPNE without the mutation.
4. If the authors are unable to perform organ on a chip assay, they can work with TCGA clinical sample data and analyze them for the newfindings suggested in the discussion, such as impact on fatty acid synthesis, lipid peroxidation
4. Please italicize the word in vitro
Reviewer 2 Report
In this revision, the authors added several experiments using HPNE/K-ras cells as well as AsPC-1 cells, which confirmed that the growth inhibition by inhibiting PLA2G2A was not PANC-1-specific phenomenon.
However, several other results were shown only in PANC-1 cells and AsPC-1 cells did not show reduced expression of fatty acid synthesis-related genes other than FASN by siPLA2G2A in Figure 4A, therefore, this the underlying mechanisms are still unclear. The authors should provide the results using AsPC-1 in Figures 3B, C, 4C, D, and 5A, B, C, D, F, otherwise, the data look like picked up arbitrarily by the authors, and I have to say that the abstract is overstating.
Previously I also commented about the side effect of PLA2G2A inhibition. The authors provided the results using HPNE/K-ras OFF cells in Figure 2B, which also showed growth inhibition to certain extent. Isn’t there any concern about the toxicity for the normal cells?
In addition, the growth inhibition in K-ras wild-type cells in Figure 2B suggests the underlying mechanisms other than oncogenic K-ras dependency. In the K-ras wild-type cells, fatty acid synthesis-related genes might also be downregulated, so, the Figure 4A-D experiments using HPNE-Kras ON/OFF cells might be informative. There might be other mechanisms than fatty acid synthesis, though.
Taken together, this revision seems a bit insufficient. The authors should perform whole experiments using at least two cell lines and should have more discussion and avoid overstating.
Round 3
Reviewer 2 Report
In this revision, the authors further added experiments using HPNE/K-ras ON/OFF cells as well as AsPC-1 cells, and now all the experiments were done using at least two cell lines. They also added discussion and tried to avoid overstating, the manuscript was improved a lot.
The authors described that FASN expression by PLA2G2A appeared independent of K-ras expression in the Discussion, and also that regulation of FASN by PLA2G2A was likely conservative and could occur regardless of K-ras status in the Results section. However, as the authors also summarized, oncogenic K-ras expression upregulates PLA2G2A expression, thereby inducing FASN expression. Interpretation of the results might be: PLA2G2A is an downstream effector regulated by oncogenic K-ras and the FASN expression is PLA2G2A-dependent.
Minor comments:
In line 175, SCD should be spelled out at the first time appearance.
In Figure 4 legend, (C) must be “Correlation between K-ras and FASN expression”.
In line 222, “driven by oncogenic K-ras” had better be deleted, since Kras-OFF cells also showed reduced expression of fatty acids synthesis-related genes.
In line 256, ROS should be spelled out and Panc-1 should be PANC-1.
There are still typos to be edited correctly.
